# Effect of Ultraviolet Light on the Shear Bond Strength of Commercial Dental Adhesives

**DOI:** 10.3390/ma18163772

**Published:** 2025-08-12

**Authors:** Markus Heyder, Stefan Kranz, Johanna Sandra Woelfel, Tabea Raabe, André Guellmar, Anna Mrozinska, Michael Gottschaldt, Ulrich S. Schubert, Bernd W. Sigusch, Markus Reise

**Affiliations:** 1Department of Conservative Dentistry and Periodontology, Jena University Hospital, An der alten Post 4, 07743 Jena, Germany; markus.heyder@med.uni-jena.de (M.H.); andre.guellmar@med.uni-jena.de (A.G.); anna.mrozinska@med.uni-jena.de (A.M.); bernd.w.sigusch@med.uni-jena.de (B.W.S.); markus.reise@med.uni-jena.de (M.R.); 2Laboratory of Organic Chemistry and Macromolecular Chemistry (IOMC), Friedrich Schiller University Jena, Humboldtstraße 10, 07743 Jena, Germany; michael.gottschaldt@uni-jena.de (M.G.); ulrich.schubert@uni-jena.de (U.S.S.); 3Jena Center for Soft Matter (JCSM), Friedrich Schiller University Jena, Philosophenweg 7, 07743 Jena, Germany

**Keywords:** adhesive dentistry, bond force, debonding on demand, orthodontic brackets, light curing, resin composites, enamel, acid etching

## Abstract

**Background:** In adhesive dentistry, debonding-on-demand is attractive for situations where no permanent attachment is required. Due to its destructive nature, ultraviolet (UV) light may be of interest for attenuating bond forces. The aim of this study was to investigate the impact of UV light on the shear bond strength (SBS) of etch-and-rinse (n = 4) and universal adhesives (n = 3). **Methods:** Glass-ceramic samples were bonded to bovine enamel surfaces (n = 10/adhesive) and subjected to shear bond testing before and after exposure to UV light (320–390 nm, 126 Jcm^−2^). Data was statistically analyzed by Mann–Whitney U test. **Results:** Initial photopolymerized etch-and-rinse adhesives showed superior SBS compared to universal adhesives. Highest values were recorded for iBOND^®^ Total etch (15.48 MPa) and Syntac classic© (17.60 MPa). Lowest SBS was obtained for Ecosite Bond^®^ (2.63 MPa). Additional UV exposure caused a significant decrease in SBS among iBOND Total etch (5.24 MPa, *p* = 0.009) and Solobond M© (3.65 MPa, *p* = 0.005), while for Syntac classic©, an increase (24.12 MPa, *p* = 0.047) was recorded. Among all other tested adhesives, no significant changes were observed. **Conclusions:** UV radiation impacted SBS of etch-and-rinse adhesives only (decrease: iBOND Total Etch, Solobond M; enhancement: Syntac classic©). Further research should focus on introducing sufficient light-triggered debonding mechanisms.

## 1. Introduction

In modern dentistry, there is a high demand for innovative and intelligent materials with adaptive characteristics to changing needs [1,2]. Since its inception in the 1950s, the development of dental adhesives has led to materials with high-quality bonding characteristics. In particular, the introduction of functional monomers, such as 4-methacryloxyethyl trimellitate anhydride and 10-methacryloyloxydecyl dihydrogen phosphate, has greatly contributed to the direct adhesion of dental materials [3,4,5].

However, in some situations where permanent attachment is not required, reversible bonding might be attractive. In particular, this is the case with the removal of orthodontic brackets or insufficient adhesive restorations, where high forces and complex grinding are usually required [6,7,8]. The technique might also be helpful in removing adhesive fiber posts.

Technically, adhesives with debonding-on-demand characteristics have been developed already. Those systems rely on a triggered attenuation of the bond force by subjection to external stimuli, such as light, heat, or electric current [9,10,11,12].

However, in dentistry, debonding-on-demand is still in the bottleneck stage. Although there are already some interesting results, the proposed systems are still far from clinical practicability. In this regard, Schenzel et al. developed an experimental adhesive that can be degraded thermally. When subjected to temperatures of 80 °C, a decrease in bond force by 94% was observed [13].

Light-triggered systems have also been developed. So far, there is one promising approach that relies on photodegradable polyrotaxanes that decompose into their molecular components after 5 min of irradiation with ultraviolet (UV) light. It was shown that UV-illumination of polyrotaxane-modified resin composite cements causes a significant decrease in tensile strength [14].

The integration of ultraviolet light-responsive moieties for a triggered attenuation of the bond force is an interesting approach, since ultraviolet photons are particularly energetic by themselves, causing chemical and structural changes among polymers already. In this regard, UV radiation can induce the breaking of covalent bonds, a process known as photolytic chain scission, which contributes to resin degradation [15,16].

Controversially, opposite effects that result from extensive exposure to UV light on adhesive materials have been reported as well. It was shown that subjection to UV radiation can result in enhanced bonding characteristics, too [17,18,19]. Also, it was found that UV exposure of titanium, stainless steel, and fiber post surfaces significantly enhances the shear bond strength of attached resin materials [20,21,22]. Nevertheless, it needs to be mentioned that UV radiation also has a significant damaging effect on living cells. It is well known that UV light can harm cells and cause malignant cell transformation. The direct interaction of UV photons with cell molecules results in DNA damage through energy absorption and in the formation of reactive intracellular radicals [23]. In the early beginnings of adhesive dentistry, UV light was used for polymerizing resin materials. Despite acceptable clinical results, limitations in the depth of cure and concerns over exposure to relatively high-intensity UV radiation caused a turnover from UV to visible light activation technology in the late 1970s and early 1980s [24].

Nevertheless, the impact of UV light on the shear bond strength (SBS) of modern dental adhesives has not yet been evaluated in detail. Currently, there are no sufficient studies available that report on the impact of UV light on the bond force of common dental adhesives. Therefore, the present in vitro study aimed to investigate the effect of an additional UV exposure on the SBS of various dental adhesives. It is assumed that additional UV illumination does not influence the bond strengths of total-etch and universal adhesives (H0).

## 2. Materials and Methods

### 2.1. Investigated Dental Adhesives

In the present in vitro study, the effect of an additional UV exposure (320 to 390 nm) on the shear bond strength (SBS) of glass-ceramic samples bonded to bovine enamel surfaces by various commercial dental adhesives was investigated. Before UV illumination, all specimens were initially light-cured using a standard light-curing unit (LCU). The manufacturer’s specifications of the respective dental adhesives are summarized in Table 1.

### 2.2. Sample Preparation

Bovine incisors (Rocholl^®^ GmbH, Eschelbronn, Germany) were embedded in SpeiFix 20 Curing Agent (Struers©, Willich, Germany) using cylindrical molds (25 × 20 mm). After embedding, the tooth samples were ground to expose the enamel surface. The enamel was then preconditioned with 35% ortho-phosphoric acid (Vococid©, VOCO©, Cuxhaven, Germany) for 15 s, afterwards rinsed (15 s) and gently dried.

Cylindrical glass-ceramic samples (Figure 1) with a diameter of 5 mm and a height of 2 mm were manufactured from IPS Empress CAD^®^ blocks (Ivoclar^®^, Schaan, Liechtenstein) and bonded to the etched enamel surface using the respective dental adhesives summarized in Table 1. All adhesives were applied according to the manufacturer’s instructions (Table 2).

Specimens were polymerized using the light-curing unit Bluephase© G2 (Ivoclar Vivadent©, Schaan, Liechtenstein) perpendicular for 30 s in direct contact with the glass-ceramic samples in full mode (1200 mW/cm^2^ ± 10%). Previously, the LCU was checked and calibrated. The main preparation steps are also visualized in Figure 1. For each dental adhesive, 20 samples were prepared.

### 2.3. UV-Light Exposure

From each group, 10 samples were subjected to UV-irradiation (test group), while the remaining 10 specimens were not exposed and served as controls. For UV illumination, the OmniCure© Series 2000 UV curing system equipped with a filter (320 to 390 nm) was used (Excelitas Technologies©, Pittsburgh, PA, USA). UV illumination was performed in a fragmented way (10 × 30 s) with breaks of 4.5 min to prevent the sample surface from heating. The output power of the UV device was set to 420 mW/cm^2^ at an iris opening of 61%. A total light fluence of 126 J/cm^2^ was delivered to each specimen. Prior to UV illumination, the device was checked and calibrated using the MARC system (Blue-Light Analytics Inc., Halifax, NS, Canada). UV irradiation was performed vertically and in close contact with the glass-ceramic samples. UV illumination was carried out in direct contact with the samples. Experiments were performed at room temperature under ambient humidity (30–60%).

### 2.4. Shear Bond Testing

Both the UV-exposed (test group, +UV-illumination) and non-exposed samples (control group, −UV illumination) were subjected to shear bond testing according to DIN EN ISO 10477. For analysis, the AllroundLine Z005^®^ testing machine (Zwick/Roell^®^, Ulm, Germany) was used. The test was carried out at a speed of 1 mm/min and a pre-load of 1 N. Total loss in attachment (Fmax) was recorded.

### 2.5. Statistical Analysis

For statistical analysis, SPSS 28.0 (IBM©, Armonk, NY, USA, Windows) was used. Due to a non-normal distribution of the obtained values, a Mann–Whitney U test was applied. An alpha of 5% was considered significant (*p* = 0.05).

## 3. Results

LCU-cured universal adhesives revealed mean shear bond strengths (SBSs) of 2.63 MPa (Ecosite Bond), 6.09 MPa (Scotchbond Universal Plus), and 3.96 MPa (Adhese Universal). In case of the etch-and-rinse adhesives, LCU-polymerization resulted in mean SBSs of 15.48 MPa (iBOND Total etch), 15.20 MPa (Prime & Bond XP), 9.94 MPa (Solobond M), and 12.1 MPa (Syntac classic). All obtained values are also listed in Table 3. As visualized in Figure 2, all etch-and-rinse adhesives showed significantly higher median shear bond values compared to the universal adhesives.

Additional subjection to UV light resulted in shear bond strengths of 1.52 MPa (Ecosite Bond), 8.63 MPa (Scotchbond Universal Plus), 5.48 MPa (Adhese Universal), 5.24 MPa (iBOND Total etch), 12.91 MPa (Prime & Bond XP), 3.65 MPa (Solobond M), and 24.12 MPa (Syntac classic). Results are summarized in Table 2.

In detail, compared to the initial LCU-based polymerization value, additional UV illumination caused a decrease in SBS by 1.11 MPa among Ecosite Bond. However, the difference was not statistically significant (*p* = 0.878). In the case of Scotchbond Universal Plus, an increase in SBS by 2.54 MPa was observed, which was also not significant (*p* = 0.333). For Adhese Universal, there was a moderate increase in the mean SBS by 1.52 MPa. Here, too, no significant differences compared to the initial LCU-based values were observed (*p* = 0.139). Results obtained from the universal adhesives are visualized in Figure 3.

Different results were obtained from the etch-and-rinse adhesives (Table 2, Figure 4). Additional UV illumination caused a significant decrease in SBS among Solobond M and iBOND Total Etch. In the case of Solobond M, a reduction by 6.29 MPa (*p* = 0.005) was observed. iBOND Total Etch showed the highest loss in SBS. Here, UV illumination caused a significant decrease by 10.24 MPa (*p* = 0.009). A drop in mean SBS was also observed for Prime & Bond XP (by 2.27 MPa), but this was not significant (*p* = 0.126).

Compared to all other tested adhesives, Syntac© classic was the only material where exposure to UV light resulted in a significant increase in shear bond strength. In comparison to the initial value, shear bond strength rose by 6.55 MPa (*p* = 0.047).

## 4. Discussion

The aim of the present in vitro study was to investigate the impact of UV radiation on the shear bond strength (SBS) of dental adhesives. As shown by the data, exposure to UV light resulted in different effects. While for iBOND Total Etch and Solobond M, a significant decrease in SBS was observed, enhanced bonding characteristics were proven for Syntac classic. Other than the observed etch-and-rinse adhesives, no significant effects were investigated among the tested universal adhesives. Therefore, H0 has to be abandoned.

However, in the present study, the respective adhesives were used for attaching glass-ceramic samples to bovine enamel surfaces. The setup was chosen because the type of glass-ceramic used did not block the transmittance of UV light, as was observed for cylindrical specimens manufactured from resin composites in a previous investigation. Since there are currently no studies available that report on the influence of UV light on the shear bond strength of common dental adhesives, the proposed experimental study is of pilot characteristics.

It was found that all photopolymerized etch-and-rinse applications were significantly more resistant to shear forces compared to the investigated universal adhesives. The highest shear bond values were recorded for iBOND Total Etch and Syntac classic. Between etch-and-rinse and universal adhesives, there is still controversy about the efficiency of enamel bonding. While some authors report superior effects of contemporary multi-bottle adhesives, others did not find any significant differences [25,26,27,28]. Anyhow, for improving the bond force of universal adhesives, additional enamel etching is advantageously [29,30,31,32]. However, in the present investigation, all tested universal adhesives still showed significantly lower shear bond values, although they were applied in the etch-and-rinse mode.

The mean SBS of the initially light-cured adhesives ranged between 2.63 MPa for Ecosite Bond and 17.60 MPa for Syntac classic. In comparison, a recent study reported shear bond values ranging between 14.5 and 21.6 MPa for Filtek Z350 resin composite samples bonded to dentin surfaces by various dental adhesives [33]. In a different study on pre-etched enamel surfaces, bond forces between 6.0 and 22.8 MPa were observed for various dental adhesives. Thus, shear bond values of 22.8 and 20.0 MPa were reported for Adhese Universal and Scotchbond Universal Plus [34]. In comparison to the results of the present study, shear bond strengths of 3.96 and 6.09 MPa were obtained for the same adhesives.

In the case of iBOND Total Etch, values of 6.96 MPa on non-etched enamel and 16.61 MPa on pre-etched dentin samples were reported in the literature [35,36]. In comparison, mean shear bond values of up to 42.9 MPa were investigated for Syntac classic on enamel samples [37]. In this regard, previous investigations of our group revealed shear bond values of 16.41 MPa for Syntac classic^®^ on pre-etched dentin samples [36]. These results are partly in line with examinations observed on bovine enamel surfaces in the present investigation.

Differences from the results of other authors might be caused by the applied method. In the present investigation, non-preconditioned (non-silanized) glass-ceramic cylinders were bonded to enamel surfaces directly. This might be a reason for the low shear bond values observed for some of the tested adhesives.

The aim of the present study was to investigate the effect of UV light on the SBS of universal and etch-and-rinse adhesives. It should be evaluated if UV radiation can induce a triggered attenuation of the SBS. As shown by the results, exposure to UV light caused a significant decrease in SBS among iBOND Total Etch and Solobond M. A loss in bond force was also analyzed for Ecosite Bond^®^ and Prime&Bond NT, which was statistically not significant. In contrast, Syntac classic revealed enhanced shear bond values. Among all tested adhesives, significant changes were only observed for the etch-and-rinse applications.

Other than what is expected, information about the impact of UV light on dental resin materials is sparse. To the best of our knowledge, there are currently no studies available that report on the effect of UV light on the bond force of commercial dental adhesives. Nonetheless, it is known that intense UV illumination can induce chemical and structural changes among polymers that have an impact on the materials’ color, shine, and surface roughness [38,39,40,41,42]. In this regard, it was proven that additional UV application (10 to 15 min) leads to increased contact angles among temporary resin composites and also results in an enhanced monomer conversion [43]. Additionally, it was reported that the use of UV light can enhance the adhesion of resin composites to enamel, which might be attributed to an increased monomer conversion and a more sufficient cross-linked polymer network [44,45].

Other authors also reported enhanced material properties after UV illumination. It was found that UV exposure of composite surfaces for 1 to 10 min significantly increased the SBS of attached samples, regardless of the resin type applied [18]. It was concluded that an increase in mechanical properties can be attributed to an enhanced polymer formation and crosslinking [44]. This could also be true for the improved shear bond values observed for Syntac classic in the present investigation.

In contrast, some authors reported on the degenerative effects that follow extensive UV illumination and are mainly based on mechanisms, such as polymeric chain scission. Degradation of the polymeric structure by UV light leads to the formation of microcracks, color changes, and the breaking of covalent bonds, which significantly affects the performance and durability of dental polymers [46,47]. Investigations using X-ray micro computed tomography additionally revealed changes in the composites’ inner structure. In response to extensive UV illumination, cracks of different shapes and volumes were found among the resin matrix, mostly in the near-surface regions [48]. UV-triggered degradation of the polymeric structure might be the reason for the observed loss in SBS among iBOND Total Etch and Solobond M in the present investigation. This is also in line with the results of a recent study that evaluated the impact of UV light on the surface layers of epoxy resins. It was found that UV radiation induces molecular chain scission, which results in weight and color changes, the formation of microcracks, and a loss in mechanical properties [49].

A more controlled light-triggered polymer degradation (debonding-on-demand) is ensured by introducing UV-sensitive moieties. This is the case with photodegradable polyrotacanes that decompose under UV illumination [14].

As also reflected by the results of the present investigation, UV illumination has different effects on dental adhesives. While for some adhesives a loss in SBS was observed, others proved enhanced bonding characteristics. Interestingly, all significant changes were detected among the tested etch-and-rinse applications only. The mechanisms behind these effects are unclear and still need to be identified.

## 5. Conclusions

It was verified that exposure to UV light (126 J/cm^2^) causes a significant decrease in SBS among iBOND Total Etch and Solobond M. At the current state, it is assumed that degeneration of the resin matrix due to polymeric chain scission and the breaking of covalent bonds is responsible for the observed attenuation in SBS. In contrast, additional exposure of Syntac classic to UV light resulted in increased shear bond values. The superior effect is likely caused by an additional monomer conversion and a more sufficient cross-linking due to UV-photon stimulation.

Ongoing examinations should focus on the identification and introduction of innovative light-sensitive mechanisms that enable a safe debonding in clinical situations where permanent adhesion is not required.

## Figures and Tables

**Figure 1 materials-18-03772-f001:**
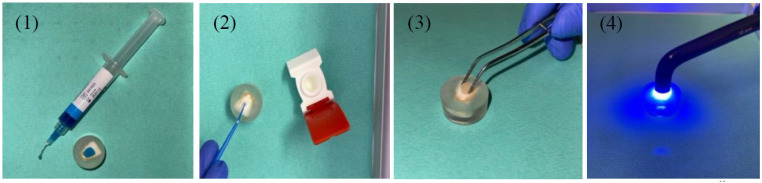
Sample preparation. (**1**) Pre-conditioning of the exposed bovine enamel surface with 35% ortho-phosphoric acid; (**2**) application of the respective dental adhesive using a microbrush; (**3**) positioning of the glass-ceramic test body; (**4**) LCU-based polymerization.

**Figure 2 materials-18-03772-f002:**
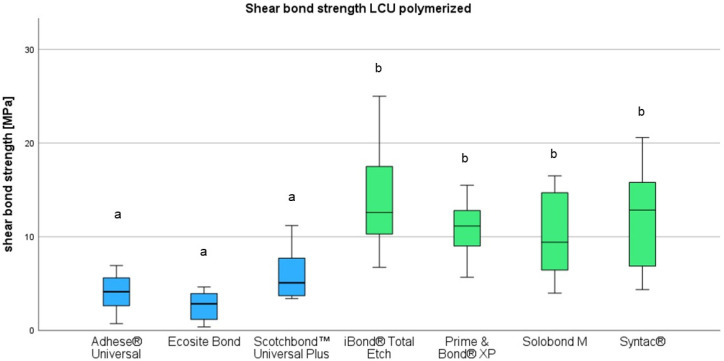
Shear bond strength of universal (blue) and etch-and-rinse (green) adhesives. Different indices indicate significant difference (*p* < 0.05).

**Figure 3 materials-18-03772-f003:**
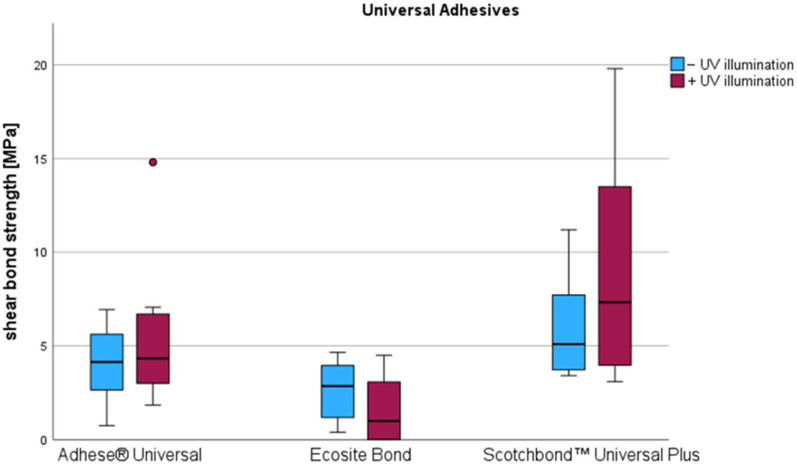
Influence of an additional UV illumination on the shear bond strength of universal adhesives. Circle indicates outlier value.

**Figure 4 materials-18-03772-f004:**
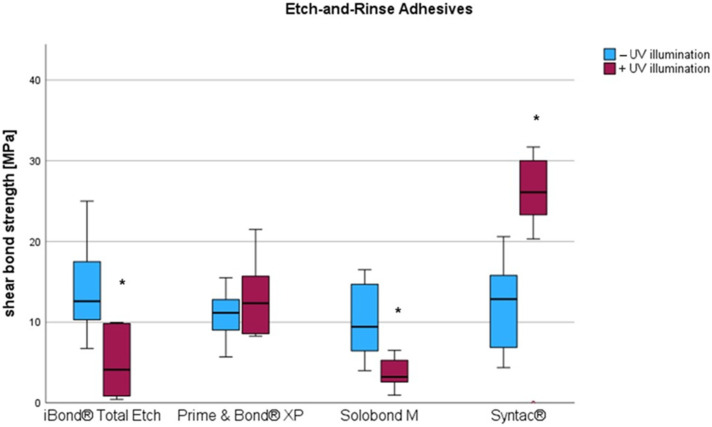
Influence of an additional UV exposure on the shear bond strength among etch-and-rinse adhesives. Significant differences between LCU polymerization (−UV illumination) and additional UV-exposure (+UV illumination) are indicated by a star.

**Table 1 materials-18-03772-t001:** Investigated dental adhesives.

	Name	Manufacturer	LOT	Ingredients
**Universal Adhesives**	Ecosite Bond^®^	DMG^®^, Hamburg, Germany	283939	HEMA, bis-GMA, ethanol, MDP, water, EHA, 2-ethylhexyl 4-(dimenthylamino)benzoate
	Scotchbond™ Universal Plus^®^	3M™^®^, Neuss, Germany	9952921	Bromilated dimethacrylate, methacrylate (HEMA), phosphorylated methacrylate, ethyl-alcohol, silanized silica, water, polymeric acid, campherquinone, aromatic amine
	Adhese^®^ Universal	Ivoclar^®^, Schaan, Liechtenstein	Z04XF0	HEMA, bis-GMA, D3MA, MDP, ethanol, water, methacrylate-modified polyacrylic acid, silicon dioxide, campherquinone, ethyl p-dimethylamino-benzoate, 2-dimethylaminoethyl-methacrylate
**Etch-and-Rinse Adhesives**	iBOND^®^ Total Etch	Kulzer^®^ Mitsui Chemicals Group, Hanau, Germany	N010608	UDMA, Ethanol, HEMA, water, methacrylated polycarboxylic acid, 4-META, glutardialdehyde
	Syntac classic©	Ivoclar^®^, Ellwangen, Germany	Primer: Z05CNM Adhesive: Z05BTZ Heliobond: Z0591M	PEG-dimethylacrylate, glutaraldehyde, water, acetone, TEGDEMA, maleic acid, bis-GMA
	Solobond M©	Voco^®^, Cuxhaven, Germany	2324174	Acetone, bis-GMA, HEMA, catalyst, BHT, acid adhesive monomer
	Prime and Bond XP©	Dentsply Sirona^®^, Charlotte, NC, USA	2302000793	TCB-resin, PENTA, UDMA, TEGDEMA, HEMA, butylated benzenediol, campherquinone, acetone, water

**Table 2 materials-18-03772-t002:** Adhesive application procedures.

	Adhesive	Application Procedure	LCU-PolymeriZation Time
**Universal Adhesives**	Ecosite Bond^®^	application time 10 s exposure time 20 s air-drying 5 s	10 s
	Scotchbond™ Universal Plus^®^	comprehensive application exposure time 20 s air-drying 5 s	10 s
	Adhese^®^ Universal	comprehensive application exposure time 20 s gently air-drying	10 s
**Etch-and-Rinse Adhesives**	iBOND^®^ Total Etch	active application exposure time 15 s air-drying 5 s	20 s
	Syntac classic©	Syntac Primer: active application, exposure time 15 s, air-drying Syntac Adhesive: active application, exposure time 10 s, sufficient air drying Heliobond: application, exposure, air-blowing to thin layer	Heliobond: 10 s
	Solobond M©	comprehensive application exposure time 30 s gently air-drying	20 s
	Prime and Bond XP©	comprehensive application exposure time 20 s air-drying 5 s	10 s

**Table 3 materials-18-03772-t003:** Mean shear bond strength (SBS) after LCU-polymerization and additional UV-exposure ± standard deviation. Significant values are marked in bold.

	Universal Adhesives			Etch-and-Rinse Adhesives			
	Ecosite Bond^®^	Scotchbond™ Universal Plus^®^	Adhese^®^ Universal	iBOND^®^ Total Etch	Prime & Bond XP©	Solobond M©	Syntac© classic
mean SBS LCU-polymerization [MPa]	2.63 ± 1.47	6.09 ± 2.63	3.96 ± 2.16	15.48 ± 8.35	15.20 ± 4.70	9.94 ± 4.53	17.60 ± 5.38
mean SBS post UV-exposure [MPa]	1.52 ± 1.59	8.63 ± 5.55	5.48 ± 3.76	5.24 ± 4.24	12.91 ± 4.27	3.65 ± 1.89	24.12 ± 9.09
mean change in SBS [MPa]	−1.11 ± 2.03	+2.54 ± 6.2	+1.52 ± 3.89	−10.24 ± 9.50	−2.27 ± 4.84	−6.29 ± 3.24	+6.55 ± 10.24
*p*-value	0.878	0.333	0.139	**0.009**	0.126	**0.005**	**0.047**

## Data Availability

The original contributions presented in this study are included in the article material. Further inquiries can be directed to the corresponding author.

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
