# Peer review of "Effect of Ultraviolet Light on the Shear Bond Strength of Commercial Dental Adhesives"

_materials, 2025, doi:10.3390/ma18163772_

Round 1

Reviewer 1 Report

Comments and Suggestions for Authors

An interesting article, well written, my congratulations. The issue is very current and the problem is still unresolved.

I have only a few minor comments, basically thoughts on my part, because the role of a reviewer is to "always be a bit smart"
Introduction
It should be remembered that at the very beginning, light-cured materials were polymerized using UV light, just like some commercial composite materials. Only after the adverse effect of this radiation on tissues was discovered, catalysts operating in the range of visible light began to be used.
It would be worth adding something about the UV hazard
In this part, you could emphasize more what thesis you will put forward before starting your research

M&M
why did you choose this particular UV radiation range, are there any literature premises for this?
ISO 10477 from which year 2020?Dentistry — Polymer-based crown and veneering materials, Why did you use this test to examine the connection of ceramics with teeth?, but it did not go beyond the testing phase in the laboratory.

Discussion
I once dealt with the problem of debonding on demand. In my case, the use of urethane acrylates, with low TG, and heating the polymerized material samples to a high temperature caused a decrease in the bonding strength.
I would try to do an IR analysis of the bonds you tested, checking the degree of C=C bond conversion between the cured sample and the same sample after exposure to UV radiation, this could help answer the question why etch-and-rinse bonds behave this way
What are the limitations of your research and what would you like to do next?

As for articles on similar topics that might be worth including in your work, I found:
doi: 10.5005/jp-journals-10005-1593
doi: 10.4103/1305-7456.149646

Good luck with your further research!

Author Response

Dear reviewer,

thank you for your valuable time reviewing our manuscript. Please find below the responses to your comments. We tried to answer each question to the best of our knowledge. All changes among the manuscript are highlighted in yellow. Thank you for your valuable suggestions that helped us to improve the scientific appearance of our study.

Comment: It should be remembered that at the very beginning, light-cured materials were polymerized using UV light, just like some commercial composite materials. Only after the adverse effect of this radiation on tissues was discovered, catalysts operating in the range of visible light began to be used.
It would be worth adding something about the UV hazard

Answer: Thank you for this important remark. Additional information about UV hazards and some historical aspects on UV-curing in dentistry were introduced to the introduction.

Comment: Why did you choose this particular UV radiation range, are there any literature premises for this? ISO 10477 from which year 2020? Dentistry — Polymer-based crown and veneering materials, Why did you use this test to examine the connection of ceramics with teeth?, but it did not go beyond the testing phase in the laboratory.

Answer: Until now, no efficient debonding method has been developed. However, the integration of UV-cleavable linker systems is considered a promising approach (please view: US20050182148, US2007142497A1 and https://doi.org/10.1016/j.reactfunctpolym.2012.04.014). The presented data origins from a funded research project that aimed on developing UV-cleavable linker systems for dental adhesives. These are intended to improve the debonding of orthodontic ceramic brackets. For this reason, the specific ISO and UV range were chosen for testing. Up to now, we did not entirely succeed in developing a suitable adhesive system. Therefore, we have not yet been able to move on beyond the laboratory stage.

Comment: I would try to do an IR analysis of the bonds you tested, checking the degree of C=C bond conversion between the cured sample and the same sample after exposure to UV radiation, this could help answer the question why etch-and-rinse bonds behave this way. What are the limitations of your research and what would you like to do next?

Answer: Thank you for the suggestions on conducting an IR analysis. We will consider this step in ongoing examinations. This might help to draw conclusions especially on the behavior of iBond and Solobond M after intensive UV exposure. Because the founded research project aimed on developing a UV light-based debonding strategy for orthodontic ceramic brackets, we used glass ceramic samples that did not block the transmittance of UV-light. One major limitation can be seen in the lack of a suitable silanization method for bonding the glass ceramic samples to the bovine enamel surfaces. This issue was also discussed in the manuscript. Right now, we are still investigating a promising adhesive that links to the glass ceramic by itself. Data about the developed experimental adhesives will be published soon.

Comments: As for articles on similar topics that might be worth including in your work, I found:
doi: 10.5005/jp-journals-10005-1593; doi: 10.4103/1305-7456.149646.

Answer: Thank you for the proposed literature. Both references were included to the manuscript. Thank you again for your kindness reviewing our manuscript.

Reviewer 2 Report

Comments and Suggestions for Authors

Dear Authors,

Thank you for submitting your manuscript. Before considering publication of the article, several revisions are recommended:

Abstract: The purpose of the study needs to be reformulated; it is not very clear. How many samples per group or used? Which types of adhesives were most affected?

Introduction: It should be explained in more detail why it is important to control the adhesion strength of dental adhesives.

It should be detailed why reversible adhesion is important in dentistry.

What data is there in the literature regarding the influence of UV on dental adhesives? More details are needed.

The limitations of previous studies regarding the influence of UV radiation on the properties of dental adhesives should be presented.

The purpose of the study is not very clearly formulated. What is the null hypothesis?

Material and method: More technical details are needed (exact distance between UV source and sample, ambient temperature, humidity, how the intensity of the light curing lamp was checked, etc).

More details are needed regarding the UV protocol.

How were the enamel dimensions and the thickness of the adhesive material between enamel and ceramic determined?

How were the samples mounted in the testing machine (on a special holder, with a clamp?)?

How was the effective adhesion area defined?

Results: more details are needed in figures and tables, not just a series of numerical values. They need to be interpreted.

Statistical results need to be interpreted.

How did UV radiation change shear strength, and what is the significance of this change? By what percent did SBS change compared to the control? It is not very clear from the results presented.

Which changes were statistically significant, and what is their clinical significance?

Discussion: The results are repeated. The positive and negative effects, depending on the type of adhesive, should be clearly explained and compared with the data in the literature.

Are these results important for clinical practice? If so, please elaborate.

What are the limitations of this study?

What are the future directions for research?

Did the results confirm the working hypothesis that you should have chosen?

Conclusions are too general. They should refer to the most important effects found.

Author Response

Dear reviewer,

thank you for your valuable time reviewing our manuscript. Please find below the responses to your comments. We tried to answer each question to the best of our knowledge. All changes among the manuscript are highlighted in green. Thank you for your valuable suggestions that helped us to improve the scientific appearance of our study.

Comment: Abstract: The purpose of the study needs to be reformulated; it is not very clear. How many samples per group or used? Which types of adhesives were most affected?

Answer: Thank you for these important remarks. As adviced, parts of the abstract were reformulated in order to clarify the purpose of the study. Please view the green marks. Number of samples were introduced (each group n=20). As shown by the results, etch-and-rinse systems were most affected by UV illumination. No significant changes were observed among the universal adhesives.

Comment: It should be explained in more detail why it is important to control the adhesion strength of dental adhesives.

Answer: Debonding on demand might be interesting in situations where permanent attachment is not required. This is the case with the debonding of orthodontic brackets or insufficient restorations. Debonding on demand might also be helpful in removing adhesive root canal posts. These issues are now described in the introduction. Please view the green marks. Thank you for these advices.

Comment: It should be detailed why reversible adhesion is important in dentistry.

Answer: Thank you for this important remark. Reversible adhesion might be interesting in situations where permanent attachment is not required. This is the case with the debonding of orthodontic brackets, removal of insufficient restorations or adhesively attached fiber posts. These issues are now described in the introduction. Reversible adhesion is an interesting approach, which is already in technical use. This issue is also addressed in the introduction. Please view lines 41-47 of the introduction (green marks).

Comment: What data is there in the literature regarding the influence of UV on dental adhesives? More details are needed.

Answer: Thank you for this remark. As already pointed out, there are unfortunately no fundamental studies available. Up to now, the influence of UV radiation on the bond force of dental adhesives has not yet been observed. There are studies that used UV radiation for aging resin-based composites. These studies are already included in study. There is only one research group that developed a UV-cleavable adhesive cement that is referenced in the introduction.

Comment: The limitations of previous studies regarding the influence of UV radiation on the properties of dental adhesives should be presented.

Answer: Thank you for this comment. There are unfortunately no studies that observe the influence of UV radiation on the bond strength of dental adhesives. There are studies available that inform about changes in the color (dental adhesive) or other material properties of resin-based composites after intensive UV illumination (UV-aging). These studies are included. Limitations are also discussed. Please view lines 216 to 219.

Comment: The purpose of the study is not very clearly formulated. What is the null hypothesis?

Answer: Thank you for this important comment. We are very sorry for missing the H0! We included a null hypothesis to the introduction. Please view lines 80-82. The purpose of the study was also redefined. Now, the aim of the study should be clarified. Please also view lines 16-18.

Comment: More technical details are needed (exact distance between UV source and sample, ambient temperature, humidity, how the intensity of the light curing lamp was checked, etc).

Answer: UV illumination was carried out in direct contact to the samples. Experiments were performed at room temperature under ambient humidity (30-60%). Light profile and fluence rate was analyzed using the MARC Patient Simulator (Blue-. Light analytics inc., Halifax, Nova Scotia, Canada). Detailed information about these issues are now included in the M&M section. Please view section 2.2.

Comment: More details are needed regarding the UV protocol.

Answer: Thank you for this important comment. We have included further details on the UV protocol. UV illumination was performed in a fragmented way for preventing heating. Changes in the manuscript are highlighted in green. Please view in section 2.2.

Comment: How were the enamel dimensions and the thickness of the adhesive material between enamel and ceramic determined?

Answer: This is an important remark. As adviced by the manufacturer, defined volumes of each adhesives were applied using the materials` specific application method. Because of this, standardized adhesive layers were produced. The ceramic test bodies were bonded to the bovine enamel directly. Enamel thickness was not measured. In order to inform about manufacturer specifications and application methods that were applied for each adhesive, table 2 was additionally introduced.

Comment: How were the samples mounted in the testing machine (on a special holder, with a clamp?)?

Answer: Samples were mounted using a holder that is part of the testing machine. Additional picture showing the holder with an attached sample can be send on request.

Comment: How was the effective adhesion area defined?

Answer: Thank you for this important question. The attached cylindrical glass ceramic sample was 5 mm in diameter. This corresponds to an effective attachment area of 19.635 mm².

Comment: More details are needed in figures and tables, not just a series of numerical values. They need to be interpreted.

Answer: We disagree in this point. Figures were prepared using SPSS. All necessary data is presented within the diagrams and tables. Subtitles and legends are included. Results are listed sufficiently in section 3. Data is interpreted and globally discussed in section 4.

Comment: Statistical results need to be interpreted.

Answer: Thank you for this comment. Significance among the results are marked. P-values are presented. Significant results are interpreted.

Comment: How did UV radiation change shear strength, and what is the significance of this change? By what percent did SBS change compared to the control? It is not very clear from the results presented.

Answer: Thank you for this question. Significant changes in SBS are reported. Significant results are indicated. In brief, additional UV exposure caused a significant decrease in SBS among iBOND Total etch (5.24 MPa, p=0.009) and Solobond M© (3.65 MPa, p=0.005), while for Syntac classic© an increase (24.12 MPa, p=0.047) was recorded. Among all other tested adhesives, no significant changes were observed. Please also refer to section 3. The exact range of changes are not presented within the manuscript. Exact data can be send on request.

Comment: Which changes were statistically significant, and what is their clinical significance?

Answer: This is an important remark that needs to be answered in detail. Additional UV exposure caused a significant decrease in SBS among iBOND Total etch (5.24 MPa, p=0.009) and Solobond M© (3.65 MPa, p=0.005), while for Syntac classic© an increase (24.12 MPa, p=0.047) was recorded. Among all other tested adhesives, no significant changes were observed. All significant changes occurred among the observed etch-and-rinse adhesives. At the current stage it was shown that only etch-and-rinse adhesives are affected by extensive UV illumination. This issue was also introduced to the manuscript. Future research should aim on introducing light-triggered systems for a debonding-on-demand function, if necessary. The results of the present study relay on an extensive UV illumination, that is not practicable (UV safety hazard). Further research should focus on the introduction of light sensitive moieties that enables low dose UV debonding in patients.

Comment: The results are repeated. The positive and negative effects, depending on the type of adhesive, should be clearly explained and compared with the data in the literature.

Answer: Thank you for this remark. Out of a range of adhesives tested, there were significant effects only among three (iBOND Total etch, Solobond M, Syntac classic). These adhesives belong to the etch-and-rinse systems. We indicated significant changes that only occurred among the etch-and-rinse adhesives in the manuscript. Unfortunately, there are no further studies available investigating the impact of UV radiation on the shear bond strength (SBS) of dental adhesives. All results are globally discussed in regard to literature available at present.

Comment: Are these results important for clinical practice? If so, please elaborate.

Answer: Thank you for this comment. As already pointed out, it was shown that the SBS of etch-and-rinse adhesives is affected by extensive UV illumination. Future research should aim on introducing light-triggered systems for a debonding-on-demand function, if necessary. The results of the present study relay on an extensive UV illumination, that is clinically not practicable (UV safety hazard). Further research should focus on the introduction of light sensitive moieties that enable a low dose UV-triggered/ visible light-triggered debonding.

Comment: What are the limitations of this study?

Answer: Thank you for this query. In the submitted study glass ceramic samples were used for not blocking the transmittance of UV-light to the respective adhesives. One limitation can be seen in the lack of a suitable silanization method for bonding the glass ceramics to the bovine enamel surfaces. This limitation was also introduced to the manuscript.

Comment: What are the future directions for research?

Answer: As also reflected by the results of the submitted investigation, UV illumination has different effects on dental adhesives. While for some adhesives a loss in SBS was observed, others proved enhanced bonding characteristics. Interestingly, all significant changes were detected among the tested etch-and-rinse applications. The mechanisms behind these effects are still unclear and need to be identified. Further examinations should focus on identify the specific mode of action. An IR analysis might be helpful, checking the degree of C=C bond conversion between a cured sample and the same sample after additional exposure to UV radiation. Further research might also focus on the  identification and introduction of innovative light sensitive mechanisms that enable a safe debonding in clinical situations where permanent adhesion is not required.

Comment: Did the results confirm the working hypothesis that you should have chosen?

Answer: No, H0 had to be abandoned. This issue was also introduced to the manuscript. Thank you for the remark on the H0!

Comment: Conclusions are too general. They should refer to the most important effects found.

Answer: UV illumination had an impact on only three adhesives observed. This is also addressed in the conclusions. At the current state, it is assumed that degeneration of the resin matrix due to polymeric chain scission and the breaking of covalent bonds are responsible for the observed attenuation in the SBS of iBond total etch and Solobond M. In contrary, additional exposure of Syntac classic to UV radiation resulted in increased shear bond values. The superior effect is probably caused by an additional monomer conversation and a more sufficient cross-linking due to UV-photon stimulation. Both issues are the main conclusions and are presented in the manuscript.

Thank you again for your valuable time reviewing our manuscript.

Reviewer 3 Report

Comments and Suggestions for Authors

Dear Authors

The topic of the manuscript is interesting, but some revisions are necessary before it can be considered for publication. Here are some suggestions to improve it:

1- Line 89: In the text it is stated that “The samples were light-cured with the Bluephase© G2 unit (Ivoclar Vivadent©, Schaan, Liechtenstein) for 30 seconds, perpendicularly and in direct contact with the glass-ceramic surface, in full mode (1200 mW/cm² ±10%). The curing unit was verified and calibrated before use.” Polymerizing with the lamp in direct contact with the sample could distort the quality of the adhesion due to the human factor. Even minimal pressure, especially at the beginning of polymerization, could break the bonds. Please address this concern.

2- Line 82 “The enamel was then pretreated with 35% orthophosphoric acid (Vococid©, VOCO©, Cuxhaven, Germany) for 15 seconds, then rinsed and dried gently.” The authors only quantified the timing of the etching. It  should be also necessary to specify the washing and swarming times because a poorly or poorly washed off etchant and/or insufficient drying could influence the adhesive properties.

3- LIne 88: Authors also followed the instructions provided by the manufacturers for the application of each individual adhesive. However, this should then generate more materials and methods and not just one such as the one reported

4- Line 94 “Sample preparation. (1) Pre-conditioning of the exposed bovine enamel surface with 35% ortho-phosphoric acid; (2) application of the respective dental adhesive using a microbrush; (3) po- sitioning of the glass-ceramic test body; (4) LCU-based polymerization.” The blowing step that is done after brushing the bonding onto the enamel, before polymerizing it, is missing. Please specify.

5- Line 164: “In the present study, adhesives were used to bond glass-ceramic samples to bovine enamel surfaces. This configuration was chosen because the type of glass-ceramic used does not block the transmission of UV light, unlike the cylindrical resin composite samples, which in a previous study (not shown) had limited transmission.” If you can’t prove it, why do you cite it? Either cite it or remove the reference.

6- Line 214: “On the contrary, it was also proven that prolonged UV post-curing times (10 to 15 minutes) lead to increased contact angles and have a positive effect on monomer conversion”; Please better describe and specify this sentence. It is not clear.

12/44 bibliographic entries are more than 10 years old. If not strictly necessary update them.

Best regards

Author Response

Dear reviewer,

thank you for your valuable time reviewing our manuscript. Please find below the responses to your comments. We tried to answer each question to the best of our knowledge. All changes among the manuscript are highlighted in purple. Thank you again for your valuable suggestions that helped us to increase the scientific value of our study.

Comment: Polymerizing with the lamp in direct contact with the sample could distort the quality of the adhesion due to the human factor. Even minimal pressure, especially at the beginning of polymerization, could break the bonds. Please address this concern.

Answer: Thank you for this comment. We agree with the reviewer. Movements during light-curing might have an impact on the polymerization result. Nevertheless, the experiments were conducted by a trained and skilled professional. Also, there were 20 specimens tested for each adhesive which minimizes the overall error.

Comment: The authors only quantified the timing of the etching. It should be also necessary to specify the washing and swarming times because a poorly or poorly washed off etchant and/or insufficient drying could influence the adhesive properties.

Answer: Thank you for this helpful comment. The time of rinsing was the same as for etching. We included this information to the manuscript. In addition, table two was introduced for more information upon the single application steps performed among each adhesive.

Comment: Authors also followed the instructions provided by the manufacturers for the application of each individual adhesive. However, this should then generate more materials and methods and not just one such as the one reported.

Answer: We agree with the reviewer. The study comprised single and multi bottle adhesives. For example: iBond total etch was only a one bottle system, while Syntac classic is composed of a primer, adhesive and bonding agent. We therefore included an additional table summarizing the single application steps. Please view table 2. Thank you for the suggestions on summarizing the applied methods in regard to the adhesives.

Comment: Sample preparation. (1) Pre-conditioning of the exposed bovine enamel surface with 35% ortho-phosphoric acid; (2) application of the respective dental adhesive using a microbrush; (3) po- sitioning of the glass-ceramic test body; (4) LCU-based polymerization.” The blowing step that is done after brushing the bonding onto the enamel, before polymerizing it, is missing. Please specify.

Answer: Thank you for this important remark. We included table 2, which summarizes all steps including air-drying as also adviced by the manufacturer.

Comment: Line 164: “In the present study, adhesives were used to bond glass-ceramic samples to bovine enamel surfaces. This configuration was chosen because the type of glass-ceramic used does not block the transmission of UV light, unlike the cylindrical resin composite samples, which in a previous study (not shown) had limited transmission.” If you can’t prove it, why do you cite it? Either cite it or remove the reference.

Answer: The paragraph was removed.

Comment: On the contrary, it was also proven that prolonged UV post-curing times (10 to 15 minutes) lead to increased contact angles and have a positive effect on monomer conversion”; Please better describe and specify this sentence. It is not clear.

Answer: Thank you for this comment. The sentence was changed. Please view lines 232 to 234.

Comment: 12/44 bibliographic entries are more than 10 years old. If not strictly necessary update them

Answer: Both references (12 and 44) of the original submission are published this year. Due to the actuality of these two publications, we will not delete these references. Because much research about UV aging of adhesive materials were published in the early 2000s, some of the references are > 10 years of age. Unfortunately, there are no current investigations published upon the issue. This was also addressed in the manuscript.

Thank you again for your valuable time reviewing our submission.

Round 2

Reviewer 3 Report

Comments and Suggestions for Authors

Dear Authors

All my suggestions have been addressed. The manuscript has been improved and is now suitable for publication.

Best regards